# Virtual Care in Nephrology: An In-Depth Retrospective Analysis of Outcomes Using the Reset Kidney Health Model

**DOI:** 10.3390/jcm13010066

**Published:** 2023-12-22

**Authors:** Benjamin A. Fritz

**Affiliations:** Reset Kidney Health, New York, NY 10001, USA; hadlee@emergentclinical.com

**Keywords:** angiotensin-converting enzyme inhibitors (ACE inhibitor), angiotensin II receptor blockers (ARB medications), chronic kidney disease (CKD), end-stage renal disease (ESRD), estimated glomerular filtration rate (eGFR), nonsteroidal anti-inflammatory drugs (NSAIDs), primary care physician (PCP), sodium-glucose co-transporter 2 inhibitors (SGLT 2 inhibitor), urine albumin-to-creatinine ratio (UACR)

## Abstract

The advent of virtual healthcare has reshaped patient management paradigms across various medical domains. This analysis examines the potential effectiveness of treating chronic kidney disease (CKD) using Reset Kidney Health’s virtual, multidisciplinary, and integrated care approach. The pilot study concentrated on evaluating the impact of this care model on the estimated Glomerular Filtration Rate (eGFR) of CKD patients over an eight-month period. The analyses showed that a majority of patients managed with the Reset Kidney Health Model experienced stability or improvements in their kidney function, as measured by eGFR. While this pilot study has several limitations, these early results suggest the potential benefits of digital healthcare innovations in chronic disease management and provide an argument for the broader integration of virtual care strategies in healthcare systems. These initial findings could lay the groundwork for further research into effectively integrating digital healthcare in chronic disease management.

## 1. Introduction

### 1.1. Background and Rationale

Chronic kidney disease (CKD) is a progressive, irreversible disease that often remains asymptomatic until it reaches advanced stages. Patients with CKD have a higher risk of hospitalization, cardiovascular mortality, cognitive decline, anemia, and bone disorders [1].

This highly prevalent disease, affecting millions of individuals globally, remains a significant public health challenge [1]. In the United States, the traditional healthcare system has struggled to offer timely, specialized nephrology care to patients suffering from CKD. A startling 72% of patients initiate dialysis without having received prior comprehensive care from a nephrologist [2]. The current referral system dependent on Primary Care Physicians (PCPs) often occurs too late, if at all. The lack of referrals contributes to over 50% + of patients initiating emergent dialysis in the hospital [3]. Even when patients receive nephrology care, existing care models suffer significant gaps. These care gaps include, for example, extensive wait times for appointments, which often last two or three months, short and insufficient consultation times with nephrologists who are more focused on dialysis patients, and poor care coordination between nephrologists and PCPs [4,5].

Advanced CKD can lead to End Stage Renal Disease (ESRD), which is extremely costly for the payers within our healthcare system. The annual costs per commercially insured patient associated with ESRD can amount to USD 121,000 without including dialysis costs [6]. A study evaluating the potential cost benefits of managing CKD to prevent further progression to ESRD found that if the rate of Estimated Glomerular Filtration Rate (eGFR) decline were decreased by 10% for patients suffering from an eGFR of less than 60 mL/min/1.73 m^2^, the savings over a ten-year period could potentially be between USD 18 and USD 20 billion [7]. It is evident from the impact on patients and the US healthcare system that patients suffering from CKD could significantly benefit from access to multidisciplinary, comprehensive CKD care.

### 1.2. The Reset Kidney Health Model

The Reset Kidney Health model, a pioneering approach to the treatment of chronic kidney disease (CKD), integrates a multidisciplinary team, including nephrologists and renal dietitians, delivering CKD care virtually. This innovative model, outlined in Figure 1, focuses on early intervention and individualized care, offering rapid access to specialty consultations, typically within two weeks of referral. Adhering to evidence-based clinical protocols, the model emphasizes comprehensive management, including regular monitoring of key indicators like the estimated Glomerular Filtration Rate (eGFR). The overarching aim is to slow disease progression and improve patient outcomes by providing holistic, patient-centered care. 

Reset Kidney Health was founded by an internal medicine physician and an individual with CKD to transform kidney care. This virtually integrated platform provides specialized care for patients with CKD through a dedicated team of nephrologists and renal dietitians. While virtual care has long been a staple in the medical field, particularly with an increased emphasis post-COVID, no other companies in the United States currently specialize exclusively in providing virtual integrated nephrology care. 

Table 1 outlines the key features of the Reset Kidney Health care model, whose unique focus sets Reset Kidney Health apart in the landscape of nephrological services, offering a distinct approach to kidney health management.

The Reset Kidney Health specialists include nephrologists and dieticians. Nephrologists can provide specialized medical care, focusing on managing kidney function and preventing further decline. A renal dietitian offers tailored dietary advice, helping patients adjust their nutrition to support kidney health and manage related conditions like hypertension and diabetes, key factors in CKD progression [8]. Combined, these specialist visits ensure a comprehensive approach to addressing kidney disease management’s medical and lifestyle aspects. Regular monitoring by a nephrologist and dietary adjustments by a renal dietitian can lead to early identification and management of complications, crucial factors in slowing disease progression. This integrated care model fosters better patient education and engagement, empowering patients to take an active role in their health, which is vital for effective long-term disease management.

Reset Kidney Health aims to tackle the systemic gaps in CKD care effectively. Our preliminary hypothesis is that our virtual, integrated, multidisciplinary approach can slow CKD progression among CKD Stage 3 and Stage 4 patients. This pilot study aims to empirically validate these outcomes by showing changes in the Estimated Glomerular Filtration Rate (eGFR). We have consistently tracked the eGFR values of our patients to offer a longitudinal perspective on CKD progression under our care model. eGFR was chosen as the basis of this study because it is considered a comprehensive measure of renal function. eGFR accounts for age, sex, race, and serum creatinine, offering a more accurate reflection of kidney health than serum creatinine or blood urea nitrogen alone. While these markers provide valuable information, eGFR integrates them into a more holistic assessment, making it a preferred choice for evaluating the effectiveness of kidney disease interventions. This focus ensures a standardized and widely accepted metric for assessing the Reset Kidney Health Model’s impact on renal function.

## 2. Materials and Methods

### 2.1. Study Design

The present investigation is a retrospective study that utilized laboratory data collected from 37 CKD patients being treated under the Reset Kidney Health care model over an eight-month period. These patients suffered not only from CKD but also experienced a variety of other health issues, the most common of which among the study population were Hypertension (HTN), Type 2 diabetes, and Coronary Artery Disease (CAD). This evaluation was to review the effectiveness of the Reset Kidney Health care model in stabilizing or improving the Estimated Glomerular Filtration Rate (eGFR) in CKD patients at stages 3 and 4 through specialized nephrological and dietary care. 

The first data point for each patient’s eGFR is collected before their initial consultation with a nephrologist. This value is the baseline measurement against which subsequent eGFR levels are compared. After the initial consultation, patients undergo routine follow-up visits with their nephrologist, generally scheduled two to four months apart. Before each follow-up visit, patients are required to complete lab work, which includes measuring their estimated Glomerular Filtration Rate (eGFR). Blood pressure, urine albumin to creatinine ratio, and glycated hemoglobin (A1C) levels were also collected when feasible. 

### 2.2. Error Margin and Interpretation

We applied a measurement error margin of ±2 mL/min/1.73 m² to interpret the estimated Glomerular Filtration Rates (eGFR). For instance, if a patient’s lab result shows a baseline Estimated Glomerular Filtration Rate (eGFR) of 24 mL/min/1.73 m², it is considered a range of 22 to 26 mL/min/1.73 m². Future measurements are evaluated in the context of this established range. 

The estimated Glomerular Filtration Rate (eGFR), a key indicator used to assess kidney function, is calculated using a combination of several variables: the serum creatinine concentration level, age, gender, and race. We consider three variabilities to be important determinants of the eGFR:Variability in Serum Creatinine: Serum creatinine levels may vary in individuals due to factors such as muscle mass, diet, and hydration status.Laboratory Measurement Variations: Different laboratories may use distinct methods or equipment to measure serum creatinine, leading to slight variations in serum creatinine results.Limitations of the Formulas: The formulas used to calculate eGFR (like the MDRD Study equation or the CKD-EPI equation) are based on population averages and may not be accurate for individual patients, especially those who are very young, very old, extremely overweight or underweight, or have unusual muscle mass.

We consider an eGFR margin of error of ±2 mL/min/1.73 m² to be clinically significant, as it will impact clinical decision-making by the nephrologist. This threshold, selected arbitrarily by the author, is considered to reflect a clinically significant change in eGFR which is likely to influence a nephrologist’s decision-making process [9].

### 2.3. Categories of Estimated Glomerular Filtration Rate (eGFR) Changes

By utilizing the criteria shown in Table 2, the pilot study’s goal was to provide a nuanced evaluation of the impact of Reset Kidney Health’s integrated care either in terms of improved or stabilized eGFR.

### 2.4. Intervention Framework

The patients had baseline estimated Glomerular Filtration Rate (eGFR) values between 16 and 59 mL/min/1.73 m², indicating they were suffering from CKD stages 3 or 4. The patients were treated under the specialized care of a team that included a nephrologist and renal dietitian.

Patient engagement commenced with an extensive 45 min consultation with a nephrologist. This initial consultation provided the foundation for future patient–nephrologist interactions, clinical decisions, and potential referrals to a renal dietitian.

Nephrologist Evaluation and Management: Nephrologist evaluation and management were characterized by an initial comprehensive patient history-gathering process. This included a review of medical records and previous laboratory data. The medical history evaluation was followed by intensive disease management based on clinical decisions using the standardized Reset Kidney Health evidence-based protocols. Furthermore, the virtual care model allowed for a detailed exploration of the patient’s daily life, living environment, and challenges they may face daily. These are relevant factors that are often overlooked in a time-constrained physical consultation.

Renal Dietitian Intervention: Dietary adjustments play a critical role in CKD management. Recognizing this, dietician consultations are offered to all patients over the course of their treatment; however, some patients decline due to personal preference. In this pilot study, 23 of the 37 patients accepted the recommendation for specialized dietetic interventions. These patients were directed to an initial 60 min consultation with the renal dietitian, which served as an extensive review of the patient’s existing dietary habits, preferences, and limitations. Following the initial consultation, a total of up to four 30 min sessions were held over the course of a year to ensure iterative feedback and refinement of dietary recommendations based on changing health metrics and patient feedback to provide optimal nutritional support specific to the individual’s needs. 

## 3. Results

Over the course of the pilot study, the Estimated Glomerular Filtration Rates (eGFR) of 37 patients with stages 3 or 4 chronic kidney disease (CKD) were monitored as they were managed under the Reset Kidney Health integrated care model. The mean age of the patient population included in this retrospective analysis was 70 years. The study population was predominantly female (78%), with only eight evaluated males (Table 3). The average length of treatment for each patient in the study was approximately four months (Table 3), during which approximately 30% of the evaluated patients experienced an improvement in the CKD stage. Not all patients were started on the care model at the same time. The first patient chosen for observation has been under the care model for 8 months, while the most recent patient has been in the program for only 2.5 months. It was observed that although there were instances of visible eGFR changes within the first 2 months, the most significant positive changes were observed after the 2-month mark. The changes in the CKD stage over the course of the pilot study are shown in Table 4 both by the number of patients in each stage group at their baseline visit versus the last time their eGFR was measured and by the number of patients who experienced improvement and the number of patients experiencing a decline. 

In addition to eGFR, other important values were monitored throughout the course of patient treatment. These values included blood pressure (diastolic; BP/D and systolic; BP/S) and glycated hemoglobin (A1C) levels. Table 5 displays the overall mean baseline blood pressure and A1C values, broken up by sex. Reset Kidney Health plans to evaluate the directionality of these lab results over time in a longer-term study.

Figure 2 below shows the change in eGFR in the CKD patient study group between the first baseline reading and the last observable measurement. The graph displays that most patients (84%) experienced either an increase or stabilization in their eGFR while enrolled in the integrated care model. 

## 4. Discussion

### 4.1. Reset Kidney Health Model

Nephrologists play a crucial role in improving or stabilizing estimated Glomerular Filtration Rate (eGFR) by providing expert medical management tailored to each patient’s specific stage of kidney disease. Our specialists are adept in diagnosing and treating conditions that impact kidney function, which includes prescribing targeted medications to slow CKD progression. Patients under their care may be prescribed SGLT 2 inhibitors, ACE inhibitors/ARBs, or Finerenone, depending on their needs. Furthermore, the Reset model emphasizes the importance of comprehensive care management, which includes vigilant blood pressure monitoring, rigorous glycemic control, statin therapy, and NSAID avoidance. This ensures that each aspect of the patient’s health that could impact kidney function is meticulously managed, playing a pivotal role in enhancing or maintaining eGFR levels.

Renal dietitians are integral in managing eGFR levels through tailored nutrition strategies. They provide specialized guidance on dietary adjustments specific to chronic kidney disease (CKD) stages, focusing on controlling protein intake, managing fluid and electrolyte balance, and reducing the load on the kidneys. This targeted nutritional counseling helps slow the progression of CKD, thus playing a crucial role in stabilizing and potentially improving eGFR among these patients.

### 4.2. Positive Trends

**Increase in eGFR**: 43% of patients showed an improvement in their Estimated Glomerular Filtration Rate (eGFR) measurements during the study period. This suggests the potential benefits of the specialized care offered by Reset Kidney Health. A sustained increase in eGFR will likely translate to a better quality of life and lower healthcare costs over time.

**Stabilization of eGFR**: In addition to those who saw improvements, 41% of the patients maintained a stable eGFR. A stable eGFR is a favorable outcome, particularly for chronic kidney disease (CKD) patients who often face a gradual decline in kidney function over time. Stabilization may be an early sign that further progression of the disease has been arrested, which is a noteworthy achievement in CKD care.

Together, these two groups represent 84% of the total study population who experienced either an increase or a stabilization in their kidney function. 

**Decline in eGFR**: Despite the positive trends observed in the majority of patients in the Reset Kidney model, 16% of the patients in this sample experienced a decline in their eGFR levels. Therefore, the Reset Kidney model was not universally effective in stabilization or improvement in eGFR. This outcome emphasizes the need for continuous improvement based on individual patient care needs.

Among the patients who consulted a renal dietitian, 50% showed improved eGFR. Conversely, only 33% of the patients who did not see a dietitian showed eGFR improvement, underscoring the potential benefits of dietary counseling in kidney care.

### 4.3. Summary and Implications

The data suggests that the Reset Kidney Health care model can be effective as 84% of the patients experienced early favorable outcomes, either in terms of improved or stabilized Estimated Glomerular Filtration Rate (eGFR), likely translating to improvement or stabilization of kidney function. This indicates the positive impact of a virtual, multidisciplinary, integrated approach to chronic kidney disease (CKD) care in which timely specialist consultations, nutritional guidance, and adherence to well-defined clinical protocols are prioritized.

The results also highlight the challenges and complexities of CKD care. The minority of patients who suffered declining eGFR (16%) raises questions for future research and suggests the need for potential modifications to the evolving care model. This retrospective analysis further adds to the existing evidence in the growing body of research advocating for integrated, specialized care for CKD patients while also emphasizing the need for ongoing studies to refine and enhance treatment methodologies.

### 4.4. Clinical Implications

Our results serve to emphasize the positive outcomes achievable when chronic kidney disease (CKD) patients are engaged in a holistic, patient-centric model of care through the Reset Kidney Health care model. This study has important implications for reducing healthcare costs, such as those associated with dialysis or hospitalization. 

### 4.5. Limitations and Future Research

The study, while promising, does have limitations that should be acknowledged:

**Sample Size**: With only 37 patients, the sample size is relatively small, which limits the generalizability of the findings.

**Short Time frame**: An eight-month period is a short window to evaluate the long-term efficacy of a CKD treatment model.

**Retrospective Nature**: Being a retrospective study, it lacks the control and randomization that are the hallmarks of a prospective randomized control trial.

**Other Important Clinical Indicators**: The current pilot study did not track Urine Albumin Creatinine Ratio (uACR), which is an important indicator of kidney damage used in guiding treatment, and in combination with eGFR, is an important prognostic indicator. This is because the majority of patients did not have a baseline uACR. In a more extensive longitudinal study, this will be tracked.

**Margin of Error**: We arbitrarily selected the eGFR margin of error to be ±2 mL/min/1.73 m² to track significant changes, which may limit the applicability of our findings. It is possible that changes in eGFR may be more evident with longer follow-up [10]. 

**Limitations of the Formulas**: The MDRD formula used to calculate eGFR in our study population is based on population averages and may not be accurate for individual patients, especially those who are very young, very old, extremely overweight or underweight, or have unusual muscle mass.

### 4.6. Plans for Continued Study

**Expanding the Sample Size**: In future studies, Reset Kidney aims to include a more extensive sample of patients to enhance the robustness and generalizability of our findings.

**Longitudinal Study**: Reset Kidney is planning a longer-term study spanning over 12 to 24 months to gain deeper insights into the sustainability of the observed improvements or stabilizations in the Estimated Glomerular Filtration Rate (eGFR). A longer-term study will also allow for the evaluation of the effects of the integrated care model on blood pressure and glycated hemoglobin (A1C).

**Multivariate Analysis**: Additional variables, such as age, comorbidities, duration of chronic kidney disease (CKD) diagnosis, use of Sodium-glucose co-transporter-2 (SGLT2) inhibitors/Angiotensin-converting-enzymes (ACEI)/Angiotensin receptor blocker (ARB), and urine albumin excretion rates will be introduced to better understand the nuances affecting the treatment outcomes.

**Establish a Control Group**: To assess the real-world effectiveness of virtual care, we will establish control groups, including those to specifically evaluate the impact of a renal dietitian.

## 5. Conclusions

The findings of this pilot study are encouraging, marking the beginning of an extensive research effort to thoroughly assess and consistently enhance the Reset Kidney Health care approach.

The early positive outcomes in Estimated Glomerular Filtration Rate (eGFR) for most patients, which include either stabilization or increase in eGFR levels, provide an impetus for extending the research to include a broader and more diverse patient pool and to explore the long-term benefits and adaptability of this virtual multidisciplinary care approach. 

## Figures and Tables

**Figure 1 jcm-13-00066-f001:**
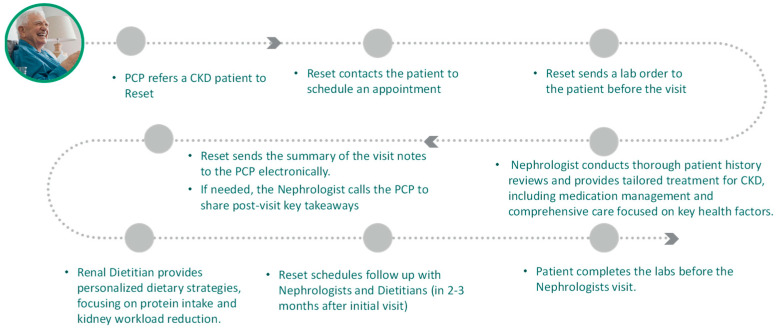
The Reset Kidney Care Model.

**Figure 2 jcm-13-00066-f002:**
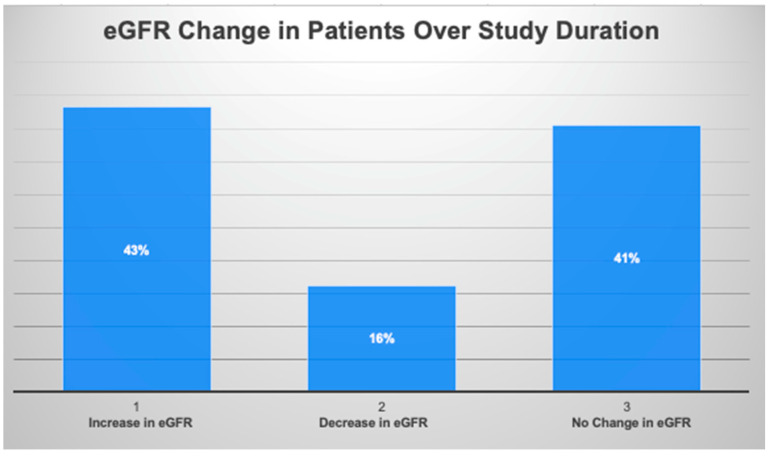
Change in Glomerular Filtration Rate (eGFR) in chronic kidney disease (CKD) patients throughout the eight-month study duration.

**Table 1 jcm-13-00066-t001:** Key Features of Reset Kidney Health’s Care Model.

Key Features	Overview
Quick Access to Specialty Care	Provides patients with the ability to secure a virtual appointment with a nephrologist within 5–10 business days.
2. A Multidisciplinary Team	Patients are offered supplemental care from a renal dietitian, focusing on the adoption of a kidney-friendly diet.
3. Standardized Care Protocols	Reset’s nephrologists adhere to a comprehensive checklist that aligns with evidence-based clinical protocols. To slow CKD progression, the patient is prescribed SGLT 2 inhibitors, ACE inhibitor/ARBs, and Finerenone as indicated. The Reset Kidney Health care model also focuses on blood pressure monitoring, glycemic control, statin therapy, and NSAID avoidance.
4. Patient Engagement	Reset Kidney Health care providers are dedicated to establishing early and trusted relationships with their patients through empathy and shared decision-making. Reset Kidney Health fosters robust patient engagement, ensuring that each patient is confident in their understanding of their disease and the care they can provide.
5. Primary Care Physician Coordination	Reset collaborates closely with the patient’s PCP to ensure seamless, comprehensive care.

**Table 2 jcm-13-00066-t002:** Estimated Glomerular Filtration Rate (eGFR) Change Criteria.

**Stable eGFR**	If a subsequent estimated Glomerular Filtration Rate (eGFR) measurement falls within the range established by the baseline (±2 mL/min/1.73 m²), it is classified as a stable GFR. For example, if the baseline eGFR is 24 mL/min/1.73 m² (range of 22 to 26 mL/min/1.73 m²), and a subsequent eGFR measurement is 25 mL/min/1.73 m², that value would be evaluated against the initial value with a standard error range of ±2 mL/min/1.73 m². Because 25 mL/min/1.73 m² falls between 22 and 26 mL/min/1.73 m², the patient’s eGFR would be considered stable between the two visits.
**Increase in eGFR:**	Any rise in the eGFR value by more than 2 mL/min/1.73 m² beyond the baseline range’s upper limit is considered an eGFR increase.
**Reduction in eGFR:**	Conversely, any decline in eGFR value by more than 2 mL/min/1.73 m² below the lower limit of the baseline range will be interpreted as a reduction in eGFR.

**Table 3 jcm-13-00066-t003:** Study Group Demographics.

Factors	Intervention Group (n = 37)
Mean Age	70 years
Gender	
Female	78%
Male	22%
Race Breakdown	
African American	14%
Asian	3%
Caucasian	84%
Dietician Recommendation Accepted	23
Mean Length of Treatment	114 days

**Table 4 jcm-13-00066-t004:** Glomerular Filtration Rate (eGFR) and chronic kidney disease staging in patients managed under the Reset Kidney Health integrated care model.

Factors	Intervention Group (n = 37)
The baseline mean of eGFR	37.69
Mean eGFR at Q3	40.79
Mean eGFR difference from baseline	+3.09
Baseline CKD Stage	
3a	9
3b	19
4	9
5	0
Last Observable CKD Staging	
2	3
3a	11
3b	16
4	7
5	0
Changes in CKD Stage	
Improved by one stage	11
Declined by one stage	1
No change in stage	25

**Table 5 jcm-13-00066-t005:** Average initial observable lab results of the chronic kidney disease (CKD) patients participating in the Reset Kidney Health care model.

**Mean BP/S**	
Overall	135.0
Male	136.2
Female	133.9
**Mean BP/D**	
Overall	76.4
Male	75.2
Female	77.7
**Average A1C**	
Overall	6.6
Male	6.6
Female	6.6

## Data Availability

Data is unavailable due to privacy restrictions.

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
