# Peer review of "Virtual Care in Nephrology: An In-Depth Retrospective Analysis of Outcomes Using the Reset Kidney Health Model"

_jcm, 2023, doi:10.3390/jcm13010066_

Round 1
Reviewer 1 Report
Comments and Suggestions for Authors
Dear Benjamin A. Fritz, thank you for interest approach. The scientific article is of undoubted interest due to the limited ability of patients to receive nephrological care in time. Overall, assessing the work positively, I kindly ask you to provide a visual diagram-drawing of what the Reset Kidney Health Model presented looks like. There are three questions that I also ask you to clarify. 1. Line 50: Is the appointment with a nephrologists virtual or face-to-face? 2. In the materials and methods, please, indicate the number of patients included in the analysis, if possible, the primary nephrological disease, and the duration of the study. These data are presented in the discussion and limitations, but it seems appropriate to present them in the materials and methods section. 3. I ask you to present the reasons for success in increasing the eGFR and it's the positive dynamics. You may probably give some practical advice to readers? It seems to me the number of literary sources cited need to be increased.
Reviewer 2 Report
Comments and Suggestions for Authors
Manuscript Title: Virtual Care in Nephrology: An In-depth Retrospective Analysis of Outcomes Using the Reset Kidney Health Model
Comments:
The author did a great job with the study titled “An In-depth Retrospective Analysis of Outcomes Using the Reset Kidney Health Model”!! I congratulate him. The author has suggested “In Chronic Kidney Disease (CKD), explored the potential effectiveness of virtual care for CKD, leveraging the unique model of Reset Kidney Health”. This can be viewed as the need for embracing digital innovation in chronic disease management.
Significance: Due to the lack of effective approaches to prevent or reverse the condition of CKD, the publication is very innovative and is an indication for further research on the model discussed. Your work significantly advances our knowledge of the information and guiding hand for the clinical practice of nephrology in CKD patients.
Limitation: Sex distribution (only 20% male in this study) and a larger sample size of CKD patients would increase the statistical power and validity of the model used in the study.
The study presents a meaningful and exciting investigation of the topic. They are advised to respond to all the comments.
Comments
Abbreviations and Acronyms
Ensure that all abbreviations are defined upon initial use.
Abstract
A slight disparity in describing purpose in the abstract can leave readers bewildered.
It should be structured in Background, Methods, Results and Conclusion.
Introduction
The author in the starting should clarify that about which population he is discussing about.
Line 30-31 “Even when patients receive nephrology care, existing care models suffer significant gaps” Reference should be cited.
Line 35-36 “Advanced CKD….. within our healthcare system”. Which healthcare system he is mentioning. Also reference 4 and 5 are too old. Recent ones should be cited.
In the Reset Kidney Health Model section, a table for stating Key Features of Reset Kidney Heath should be included for adding scholarship in this area.
No need of mentioning Hypothesis and Study Aim section separately. It can be listed a s last para of introduction. Consider restructuring the content.
Kindly clarify why only eGFR is considered as established measure of renal function, whereas Serum Creatinine, Serum Blood Urea Nitrogen and GFR should also be considered for validating the model.
Whether the Reset Kidney Health Model has been used elsewhere or not should be mentioned.
Importance of Reset Health Kidney Model should be written in a sufficiently deepened way for clearer understanding.
Methods
The author has mentioned in line 75 “The main objective was to……..”. It should be mentioned with hypothesis and aim section. Further the meaning of aim and objective of the study is similar. Kindly modify either of the one for better meaning.
Categories of GFR changes can also be listed as table.
Special Diet followed by the patients should be mentioned.
Results
Table 1, in the first row 69 years old, “old” should be removed. Also, in the dietary recommendation section, 12 patients, “patients” should be removed.
A comparison between the two approaches among the dietary interventions and its effect on eGFR will highlight the significance and pragmatic clinical implications of the discussed topic.
Figure 1 should be more explanatory, as the trend of eGFR is not explaining, when the model was implicated, what was the baseline and others.
Discussion
Improve the section for logical continuation of text. Flow chart explaining the role of each component of the model in increase or stabilization of eGFR will strengthen the discussion.
References
Should be more in number, recent and relevant to the research and should follow a common citing style.
Minor editing of the English language is needed.
Comments on the Quality of English Language
Minor editing.
Reviewer 3 Report
Comments and Suggestions for Authors
Dear authors - You have addressed a critically important topic with respect to CKD. There is potentially very good information in what your retrospective study has found, but more detail is needed.
1. Please define the time frame for your retrospective analysis. A minimum of two months is mentioned, but what what was the intent? Change in creatinine over a specific time? A year? Longer? Is 154 days in Table 1 the mean period of follow-up?
2. The title of your manuscript suggests the visits are telehealth. Were all visits, with all members of the multi-disciplinary team virtual?
3. Do you have data on patient adherence?
4. Table 1: Typically do not need to add both female and male, unless you also need to account for non-binary subjects. Why were only 12 of 30 patients recommended for dietary consultation. Dietary counseling should probably be instituted for all patients with CKD 3 or worse.
5. How many Nephrologists were involved? Though you had standardized protocols, do you know how closely they were followed?
7. Table 2 - I don't find the information about the CKD stage at baseline and after "treatment" that informative. It may be more helpful to present and average change in eGFR before and after as a line plot with whiskers for variation
8. Do you have information about which interventions were implemented and the change in eGFR? You cannot attribute cause and effect with a retrospective study but maybe an association could be seen, though your n of 30 is small
9. Table 3 really does not add value. This could all be said in the text of the manuscript.
10. This is an important topic and the discussion just reiterates some of your results for the most part. This is the opportunity to emphasize the importance of early intervention at a multi-disciplinary level and a call for prospective monitoring. It would be important to emphasize how your Reset Model might be implemented.
Round 2
Reviewer 2 Report
Comments and Suggestions for Authors
No further comments
Author Response
N/A due to no further comments